# A digital microscope for the diagnosis of *Plasmodium falciparum* and *Plasmodium vivax*, including *P. falciparum* with *hrp2/hrp3* deletion

Yalemwork Ewnetu[1,2], Kingsley Badu [3], Lise Carlier[4], Claudia A. Vera-Arias[5], Emma V Troth [5], Abdul-Hakim Mutala [3], Stephen Opoku Afriyie [3], Thomas Kwame Addison[3], Nega Berhane[1], Wossenseged Lemma[6], Cristian Koepfli [5]*

**1** Department of Medical Biotechnology, Institute of Biotechnology, University of Gondar, Gondar, Ethiopia, **2** University of Gondar Comprehensive Specialized Hospital, Gondar, Ethiopia, **3** Department of Theoretical and Applied Biology, Kwame Nkrumah University of Science and Technology, Kumasi, Ghana, **4** LMC Projects, Amsterdam, The Netherlands, **5** Department of Biological Sciences & Eck Institute for Global Health, University of Notre Dame, Notre Dame, Indiana, United States of America, **6** Department of Medical Parasitology, School of Biomedical and Laboratory Sciences, Collage of Medicine and Health Sciences, University of Gondar, Gondar, Ethiopia

* ckoepfli@nd.edu

**Data Availability Statement:** All data is available as S1 Data.

## Abstract

Sensitive and accurate malaria diagnosis is required for case management to accelerate control efforts. Diagnosis is particularly challenging where multiple *Plasmodium* species are endemic, and where *P. falciparum hrp2/3* deletions are frequent. The Noul miLab is a fully automated portable digital microscope that prepares a blood film from a droplet of blood, followed by staining and detection of parasites by an algorithm. Infected red blood cells are displayed on the screen of the instrument. Time-to-result is approximately 20 minutes, with less than two minutes hands-on time. We evaluated the miLab among 659 suspected malaria patients in Gondar, Ethiopia, where *P. falciparum* and *P. vivax* are endemic, and the frequency of *hrp2/3* deletions is high, and 991 patients in Ghana, where *P. falciparum* transmission is intense. Across both countries combined, the sensitivity of the miLab for *P. falciparum* was 94.3% at densities >200 parasites/µL by qPCR, and 83% at densities >20 parasites/µL. The miLab was more sensitive than local microscopy, and comparable to RDT. In Ethiopia, the miLab diagnosed 51/52 (98.1%) of *P. falciparum* infections with *hrp2* deletion at densities >20 parasites/µL. Specificity of the miLab was 94.0%. For *P. vivax* diagnosis in Ethiopia, the sensitivity of the miLab was 97.0% at densities >200 parasites/µL (RDT: 76.8%, microscopy: 67.0%), 93.9% at densities >20 parasites/µL, and specificity was 97.6%. In Ethiopia, where *P. falciparum* and *P. vivax* were frequent, the miLab assigned the wrong species to 15/195 mono-infections at densities >20 parasites/µL by qPCR, and identified only 5/18 mixed-species infections correctly. In conclusion, the miLab was more sensitive than microscopy and thus is a valuable addition to the toolkit for malaria diagnosis, particularly for areas with high frequencies of *hrp2/3* deletions.

**Funding:** This work was funded by Noul Inc, the developer of the miLab digital microscope, through grants to CK. The authors have discussed the results with Noul throughout the project, and taken the joint decision to publish. It was the sole responsibility of the corresponding author to ensure all data is accurate, and that data interpretation is correct.

**Competing interests:** This work was funded by Noul Inc, the developer of the miLab digital microscope. Noul Inc. holds all intellectual property of the miLab device. Noul awarded grants to the University of Notre Dame for CK to conduct the research. Noul also provided funding to LMC Projects for LC to conduct research, and to WL and YE to compensate them for expenses related to the research, including per diem payments. None of the authors holds any intellectual property rights in the miLab device, or shares of Noul. The authors have discussed the results with Noul throughout the project, and taken the joint decision to publish. It was the sole responsibility of the corresponding author to ensure all data is accurate, and that data interpretation is correct. The relationship between the authors and Noul did not alter their adherence to PLOS policies on sharing data and materials. There are no patents, products in development or marketed products associated with this research to declare. All commercial affiliations for all authors are included in the manuscript.

# Background

Malaria remains a global health threat causing over 600,000 deaths and nearly 250 million clinical cases in 2021 [1]. Diagnosis of clinical cases presenting to health centers is typically done by light microscopy, or by rapid diagnostic test (RDT). The sensitivity of these tools is limited; diagnosis is particularly challenging when multiple parasite species co-circulate in a population, for example *Plasmodium falciparum* and *Plasmodium vivax*.

The sensitivity and specificity of diagnosis by microscopy depends heavily on the expertise of the microscopist. Field microscopists typically do not detect infection at densities of less than 100–200 parasites per μL blood. They detect a much lower proportion of infections than WHO–certified expert microscopists [2,3]. In the last two decades, RDTs have become a common alternative to microscopy. RDTs are lateral flow devices that detect parasite-specific proteins through immunohistochemistry. RDTs require minimal infrastructure, and results are available within approximately 20 minutes.

The most sensitive RDTs for the diagnosis of *P. falciparum* detect the Histidine Rich Protein 2 (HRP2). These RDTs also detect HRP3 which shares a highly similar sequence with HRP2. HRP2 is expressed abundantly by the parasite and released to the plasma and thus an ideal target for diagnosis. HRP2-based RDTs are more sensitive than other RDTs that detect, for example, parasite Lactate Dehydrogenase (pLDH) or Aldolase [4,5]. In 2010, reports from Peru first described *P. falciparum* strains lacking the *hrp2* gene [6]. Since then, *hrp2* negative parasites have been reported from a number of countries (reviewed in [6]), raising concerns that the currently used HRP2-based RDTs may substantially underdiagnose *P. falciparum*. The WHO recommends switching to alternative diagnostics if >5% of parasites carry *hrp2* deletions [8]. RDTs can yield false-positive results if the antigen persists after parasite clearance, and several infectious and non-infectious diseases can lead to false-positive RDTs [9–11]. As a result of limited sensitivity of RDTs, *hrp2/3* deletions, and possible false-positive results, visual observation of parasites under a microscope provides greatest confidence in the diagnosis.

Automated microscopy using artificial intelligence (AI) to distinguish between infected and non-infected red blood cells (RBCs) in blood smears is a promising new approach for malaria diagnosis [12]. Different automated slide readers are being developed and tested [14,14]. These readers typically require manual preparation of the blood film and staining before automated detection. These algorithms showed sensitivity similar to microscopy at health centers [14]. The miLab, developed by Noul, is a fully integrated digital microscope for malaria diagnosis. A droplet of blood is added to a cartridge, which is then inserted into the miLab device. The device conducts blood film preparation including smear and Giemsa staining, and automated image analysis [16,17]. An algorithm detects infected RBCs, and displays the species detected, and parasite density. RBCs are shown on the display, allowing the operator to confirm results obtained by the algorithm (including 10 images from each cell along the Z-axis). Time-to-result is approximately 20 minutes, with less than two minutes hands-on time. The miLab device is portable with a weight of approximately 11 kg (Fig 1). It can run on a built-in battery for three to four hours in case of a power outage.

We tested the miLab for the diagnosis of clinical malaria infections in Ethiopia and Ghana. In Gondar in the Ethiopian highlands *P. falciparum* and *P. vivax* are endemic, and a large proportion of *P. falciparum* infections carry *hrp2/3* deletion [18], presenting challenges for diagnosis. In Ashanti region of Ghana, transmission of *P. falciparum* is intense [19]. Across both sites, we collected 1649 blood samples from febrile patients, and compared diagnosis by miLab to local microscopy, expert microscopy, RDT, and qPCR.

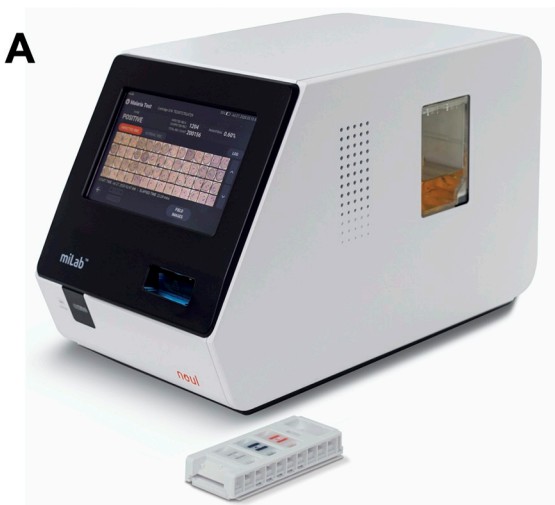
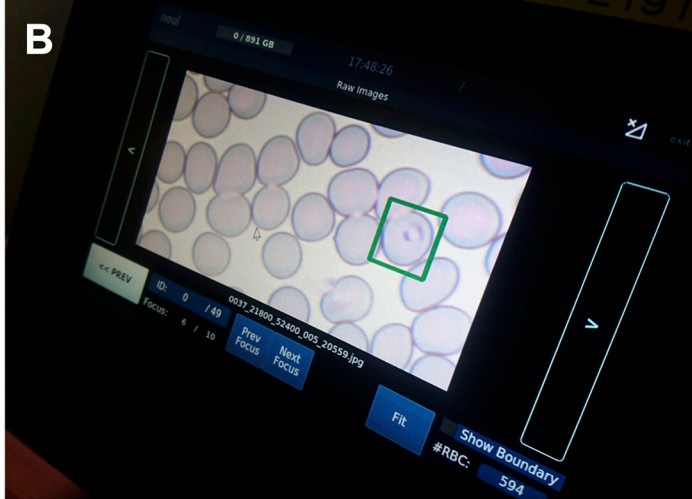

**Fig 1. miLab device.** A) Device with cartridge in front. B) Infected RBC detected by the algorithm and displayed on screen.

## Study population and methods

### Ethical approval

Informed written consent was collected from each individual, or, in the case of minors, from the legal guardian prior to sample collection. Ethical approval was obtained from University of Gondar Vice President of Research & Community Service (approval no. V/P/RCS/05/44/2019), the Kwame Nkrumah University of Science and Technology College of Health Sciences Committee on Human Research, Publication, and Ethics (approval no. CHRPE/AP/573/19), and the University of Notre Dame Institutional Review Board (approvals no. 19-08-5511, 19-04-5321).

### Study sites and sample collection

In Ethiopia, the study was conducted in Maraki health center in the outskirts of Gondar. Gondar is a town of about 200,000 people located at approximately 2,100 meters above sea level. Malaria transmission is perennial, with a major peak from October to January, and a minor peak around May to July. Samples were collected during the main transmission season between November 1, 2021 and January 31, 2022. All individuals above one year of age and referred for testing for malaria were eligible to participate. In addition to blood collected for routine diagnosis by microscopy, approximately 150 μL blood was collected for diagnosis by miLab, RDT, and qPCR. All samples were tested on site by miLab, and RDT, and the result from the diagnosis conducted routinely by the local microscopist at the health center was recorded. In Ethiopia, samples were tested using the CareStart Malaria Pf/Pv(HRP2/pLDH) Ag combo RDT (lot MO19H71). At the time of the study, this test was used by the malaria control program. For qPCR, 50 μL blood was spotted on filter paper, air dried, packed into individual zip-lock bags, and stored at -20˚C until extraction.

In Ghana, the study was conducted in Agona and Mankranso Government Hospitals near Kumasi where transmission is perennial. Samples were collected from January 26 to August 10, 2021. As in Ethiopia, all individuals above one year of age and referred for testing for malaria were eligible to participate, and an additional approximately 150 μL of blood was collected for this study. In Ghana, samples were screened with the Rapigen BIOCREDIT Malaria

Ag Pf (pLDH/HRPII) RDT. This novel test has bands for HRP2 and LDH, and is more sensitive than RDTs that had been available previously (including the RDT used in Ethiopia) [5]. Expert microscopy has been considered the gold standard for parasite diagnosis and quantification for many decades [20]. As the protocol follows strict guidelines, expert microscopy data from different studies can be directly compared [20,22]. For expert microscopy, thick and thin blood films were prepared and read by two expert microscopists. At least 100 high power fields were examined before adjudging slides as positive or negative. Parasite quantification was estimated based on the total number of malaria parasites counted per 200 or 500 white blood cells and then multiplied assuming 8000 white blood cells per μL blood [20]. For qPCR, whole blood was kept in 2 mL EDTA microtainers at -20˚C until DNA extraction.

For miLab diagnosis in the health center in both countries, 200,000 RBCs were screened, corresponding to 0.04 μL blood. The miLab was operated on site, i.e. in the health center, either by local study personnel from University of Gondar or KNUST, or by health center personnel after a brief introduction. Operators of the miLab were not blinded, i.e., they were aware of the results of microscopy and RDT conducted at the health centers. A subset of slides from Ethiopia was further screened for 500,000 RBCs, corresponding to 0.1 μL blood. This screening was conducted in the Noul laboratory in South Korea.

### *P. falciparum* and *P. vivax* qPCR, and *hrp2/3* deletion typing

qPCR can reach a limit of detection of <1 parasite per μL blood and was conducted as most sensitive assay to detect and quantify parasites [23]. Samples (dried blood spots (DBS) from Ethiopia and whole blood in EDTA microtainers from Ghana) were shipped on ice packs to the University of Notre Dame for DNA extraction. For extraction of DBS from Ethiopia, 5 punches 3 mm in diameter were used, corresponding to approximately 15 μL of blood, and eluted into 50 μL elution buffer. For extraction of whole blood from Ghana, 100 μL of blood was used for extraction and eluted into an equal volume of elution buffer. Detailed extraction protocols have been published previously [23].

The *var*ATS qPCR assay was used to screen for *P. falciparum*, and the mitochondrial *cox1* assay for *P. vivax* [24]. DNA recovery differs substantially between extractions from DBS and whole blood [23]. To obtain comparable quantification of parasite densities, different external standards were prepared. For absolute quantification of DBS samples, parasites were quantified by digital PCR, mixed with uninfected blood at different densities, spotted on filter paper, and extracted along field samples. Using these protocols, the 95% probability of detection for *P. falciparum* is 2.3 parasites/μL blood [23], and slightly higher for *P. vivax* as 10–12 mitochondria are expected per parasite compared to approximately 20 *var*ATS copies that are amplified. For whole blood samples, a standard curve was prepared from parasites quantified by digital PCR. The 95% probability of detection for *P. falciparum* in whole blood samples is 0.3 parasites/μL blood [23]. Sensitivity of diagnostic tools was calculated at 200 and 20 parasites/μL. As these thresholds are substantially higher than the LOD of the qPCR on DBS and whole blood, different sample collection methods are not expected to affect results. Researchers conducting the qPCR were not blinded, i.e., they were aware of the results by local microscopy, expert microscopy, RDT, and miLab.

Samples from Ethiopia positive for *P. falciparum* were typed for *hrp2/3* deletions by digital PCR [24]. In brief, a probe-based assay for either *hrp2* or *hrp3* was multiplexed with an assay for a control gene (*serine-tRNA ligase*). The *hrp2* assay amplifies a fragment in exon 2 immediately upstream of the histidine repeats that are detected by the RDT. This region of the gene is deleted among all samples were breakpoints were reported. The *hrp3* assay amplifies a fragment in the center of exon 2 [24]. The reaction mix was subdivided into 8500 partitions of 0.34

nL each on the Qiagen QIAcuity instrument, followed by end-point PCR. Only partitions with DNA template for either *hrp2/hrp3* or *serine-tRNA ligase* will contain amplification product, which is measured through fluorescent emission. The number of partitions with amplification product represents the number of PCR templates in the initial sample. For a wild-type parasite with no deletion, the number of partitions with *hrp2* or *hrp3* signal will be very similar to the number of partitions with signal for *serine-tRNA ligase*. A parasite carrying the deletion will still show a signal for *serine-tRNA ligase*, but not for *hrp2* and/or *hrp3*. Samples were included in the analysis if at least 5 partitions were positive for tRNA.

## Data analysis

Limited data on the number of positive samples, and on the number of infections with densities >200 and >20 parasites/µL was available before the study. Thus, not formal sample size calculation could be conducted. The number of samples exceeds the number required to be able to detect a 5% difference in sensitivity with 95% confidence, which is 377 samples.

All data is available in S1 Data. The WHO evaluates the sensitivity of RDTs against two parasite density thresholds, 2000 and 200 parasites per µL blood. Most RDTs reach a sensitivity of well over 95% against the 2000 parasites/µL threshold, however many infections are of low density. We thus evaluated the miLab against the lower threshold of 200 parasites/µL, as well as against another 10-fold lower threshold of 20 parasites/µL based on qPCR.

The presence of *P. falciparum* and *P. vivax*, and mixed-species infections in Ethiopia warranted specific rules for data analysis. In case parasites are detected by the miLab, the device suggests a parasite species and shows infected RBCs on the screen. This allows the operator to validate and if needed to correct the species detected. To calculate the sensitivity of the miLab in Ethiopia, a sample was counted positive if the miLab detected infected RBCs for either parasite species, *P. falciparum* or *P. vivax*. If a sample was positive by qPCR for *P. falciparum* but the miLab diagnosed possible *P. vivax* (or vice versa), it was counted as positive. In line with this, calculations of specificity for the miLab method were restricted to samples negative by PCR for either species. In addition, the number of infections where the incorrect species was diagnosed, or mixed-species infections with only one species diagnosed, are presented separately.

Given the high frequency of *hrp2/3*deletions in Ethiopia, only *P. falciparum* infections without *hrp2* deletions (but irrespective of *hrp3* deletion status) were included in calculations for RDT sensitivity and specificity.

Area under the curve (AUC) was calculated in Stata v.16.1 using a nonparametric estimation of the ROC. The miLab, RDT, and microscopy are not expected to detect very low-density infections. Samples were classified as true positives if parasite density was >20 parasites/µL, and as true negatives if they were negative by qPCR. Samples with densities >0 to < = 20 parasites/µL were excluded from AUC calculations.

The miLab also determines parasite density, i.e., number of infected RBCs. Counts by miLab were compared to expert microscopy data from Ghana. Samples were excluded if the density determined independently by two expert microscopists was >2-fold. For the samples included in the analysis, the mean of the two microscopists was calculated. Density by miLab was extrapolated to parasites/µL assuming 5,000,000 RBC per µL blood.

## Results

In Ethiopia, 659 febrile patients were enrolled between November 1, 2021 and January 31, 2022. By DBS-based qPCR, 495/659 (75.1%) individuals tested positive. 279/659 (42.3%) tested positive for *P. falciparum* mono-infection, 79/659 (12.0%) for *P. vivax* mono-infection, and

**Table 1. Sensitivity and specificity for the diagnosis of *P. falciparum* and *P. vivax* compared to qPCR as gold standard.**

| A) *P. falciparum* | | | | |
|---|---|---|---|---|
| | Sensitivity at 200 parasites/µL | Sensitivity at 20 parasites/µL | Specificity | AUC >20 parasites/µL |
| | | Ethiopia | | |
| miLab | 98.0% (145/148) | 96.3% (154/160) | 86.0% (141/164) | 86.30% |
| Microscopy[1] | 83.1% (123/148) | 78.1% (125/160) | 99.4% (163/164) | 87.00% |
| RDT (HRP2)[2] | 89.5% (85/95) | 86.7% (85/98) | 100% (243/243) | 78.40% |
| | | Ghana | | |
| miLab | 91.4% (169/185) | 74.6% (188/252) | 96.3% (567/589) | 85.40% |
| Microscopy[1] | 83.8% (155/185) | 64.7% (163/252) | 98.5% (580/589) | 81.60% |
| RDT (HRP2/LDH)[3] | 94.6% (175.185) | 84.1% (212/252) | 98.0% (577/589) | 91.00% |
| | | Combined | | |
| miLab | 94.3% (314/333) | 83.0% (342/412) | 94.0% (708/753) | 86.30% |
| Microscopy1 | 41.6% (138/333) | 34/8% (143/411) | 93.5% (704/753) | 81.40% |
| B) *P. vivax* | | | | |
| | Sensitivity at 200 parasites/µL | Sensitivity at 20 parasites/µL | Specificity | AUC >20 parasites/µL |
| miLab | 97.0% (65/67) | 93.9% (77/82) | 97.6% (160/164) | 87.10% |
| Field microscopy | 80.1% (54/67) | 67.0% (55/82) | 97.0% (159/164) | 81.30% |
| RDT | 91.0% (61/67) | 76.8% (63/82) | 97.6% (160/164) | 87.30% |

All density thresholds are based on qPCR data. Numbers in brackets show number of positive or negative samples over total number of samples.

[1] In Ethiopia, microscopy data from the health center was recorded (i.e. no expert microscopy was conducted). In Ghana, expert microscopy data was recorded.

[2] Samples with *hrp2* deletion were excluded from calculations.

[3] Any RDT positive either for LDH or HRP2 was considered positive.

137/659 (20.8%) carried mixed infections with *P. falciparum* and *P. vivax*. In Ghana, 991 febrile patients were enrolled between January 26 and August 10, 2021. By qPCR on whole blood, 402/991 (40.6%) tested positive for *P. falciparum*.

## Sensitivity and specificity of *P. falciparum* and *P. vivax* diagnosis

Across both sites combined, sensitivity of the miLab was 94.3% at densities >200 parasites/µL (Ethiopia: 98.0%, Ghana: 91.4%, Table 1). At densities of >20 parasites/µL, sensitivity was 83.0% (Ethiopia: 96.3%, Ghana: 74.6%, Table 1). In comparison, RDT sensitivity was 89.5% in Ethiopia and 94.6% in Ghana at >200 parasites/µL, and 86.7% and 84.1% at >20 parasites/µL (Table 1). The miLab exceeded the sensitivity of microcopy conducted at the health post (conducted only in Ethiopia), as well as the sensitivity of expert microscopy (conducted only in Ghana, Table 1).

Specificity for *P. falciparum* of the miLab was 94.0% across both sites (Ethiopia: 86.0%, Ghana 96.3%). Even though samples positive for qPCR for *P. vivax* were excluded from calculations of specificity, in Ethiopia, specificity was low compared to microscopy and RDT. 23 samples were positive by miLab with 1–5 parasites detected, but negative by qPCR for either species (details on samples positive by miLab for *P. falciparum*, but for *P. vivax* only by qPCR are provided below). All 23 samples were also negative by microscopy and RDT. In Ghana, 22 samples were positive by miLab but negative by qPCR. Most were confirmed negative by RDT and microscopy, 2 were positive by expert microscopy, and 4 were positive by RDT. AUC of the miLab was similar to microscopy and RDT (Table 1).

The sensitivity of miLab for *P. vivax* was 97.0% at densities >200 parasites/µL, and 93.9% at densities >20 parasites/µL, and higher than diagnosis by microscopy or RDT with sensitivities

of 67.0–91.0% (Table 1). Only four samples were determined positive for *P. vivax* by miLab but negative for either species by qPCR, resulting in a specificity of 98.7%. All four samples were *P. vivax* positive by microscopy and/or RDT.

## Diagnosis of *hrp2/3* negative *P. falciparum* infections

In Ethiopia, 150 *P. falciparum* positive samples with densities >20 parasites/μL were successfully typed for *hrp2* deletion and 141 for *hrp3* deletion. 52/150 (34.7%) carried *hrp2* deletion, and 129/141 (91.5%) carried *hrp3* deletion. 45/140 (32.1%) carried double deletions. The miLab successfully diagnosed 51/52 (98.1%) infections with *hrp2* deletion (45 also carried *hrp3* deletions, 3 were wild type, and for 4 samples no *hrp3* deletion typing data was available). Five infections were misdiagnosed as *P. vivax*; three of them were also *P. vivax* positive by qPCR. Field microscopy diagnosed 37/52 (71.2%) of *hrp2*-negative infections.

## Accuracy to distinguish between *P. falciparum* and *P. vivax*

In Ethiopia where *P. falciparum* and *P. vivax* are frequent, a number of infections were diagnosed by the miLab, but either the wrong species was assigned, or only one species was diagnosed in mixed species infections. Including qPCR-positive infections of any density, of 209 infections determined positive for *P. falciparum* by the miLab, 13 were *P. vivax* mono-infections by qPCR (Table 2). Of 75 Infections diagnosed as *P. vivax* mono-infection by the miLab, eight were positive only for *P. falciparum* by qPCR (Table 2).

Restricting the analysis to samples with >20 parasites/μL, 9 samples were positive for *P. falciparum* by miLab, but by qPCR only *P. vivax* was detected at >20 parasites/μL (Table 2A). Six samples were positive by miLab for *P. vivax* only, yet qPCR only detected *P. falciparum* at >20 parasites/μL (Table 2A). All five infections diagnosed as *P. falciparum*/*P. vivax* mixed infections by miLab were confirmed by qPCR. An additional 13 samples with *P. falciparum* and *P. vivax* at >20 parasites/μL were diagnosed by miLab either as *P. falciparum* mono-infection (N = 5), or *P. vivax* mono-infection (N = 8) (Table 2A). Thus, among samples positive by miLab and at densities >20 parasites/μL by qPCR, the wrong species or only one of two species was assigned to 13.1% (28/213) of samples.

Microscopy, at densities >20 parasites/μL (by qPCR), misclassified 25/178 (14.0%) mono-infections among samples found positive, and correctly identified 5/17 (29.4%) mixed-species infections (Table 2B). Among samples positive by microscopy and at densities >20 parasites/μL by qPCR, the wrong species or only one of two species was assigned to 19.0% (37/195) of samples.

## Additional infections diagnosed when screening 500,000 RBCs

Using standard settings, the miLab screens 200,000 RBCs for parasites. A subset of 52 samples from Ethiopia initially negative by miLab were screened for 500,000 RBCs. Assuming 5,000,000 RBCs per microliter of blood, this larger volume corresponds to 0.1 μL blood screened. Among the 52 samples, by qPCR 21 were negative, 18 carried *P. falciparum* mono-infection, 2 carried *P. vivax* mono-infection, and 11 carried *P. falciparum*/*P. vivax* mixed infections. Screening of 500,000 RBCs resulted in 4 additional samples testing positive by miLab. Three were *P. falciparum* mono-infections with densities of 0.2, 17, and 107 parasites/μL by qPCR, and one was mixed infection with a density of 213 *P. falciparum* and 18 *P. vivax* per μL. S1 Fig shows the density distribution of samples rescreened by the miLab.

**Table 2. Agreement in species identification by miLab (A) or microscopy (B) and qPCR.**

| A) miLab | | qPCR (all densities) | | | |
|---|---|---|---|---|---|
| | Total | Pf | Pv | Pf+Pv | Negative |
| miLab negative | 370 | 146 | 33 | 54 | 137 |
| miLab detects Pf | 209 | 125 | 13 | 48 | 23 |
| miLab detects Pv | 75 | 8 | 33 | 30 | 4 |
| miLab detects Pf+Pv | 5 | 0 | 0 | 5 | 0 |
| Total | 659 | 279 | 79 | 137 | 164 |
| | | qPCR (>20 parasites/μL) | | | |
| | Total | Pf (>20 parasites/μL) | Pv (>20 parasites/μL) | Pf+Pv (>20 parasites/μL) | Negative |
| miLab negative | 11 | 6 | 5 | 0 | NA |
| miLab detects Pf | 144 | 130 | 9 | 5 | NA |
| miLab detects Pv | 64 | 6 | 50 | 8 | NA |
| miLab detects Pf+Pv | 5 | 0 | 0 | 5 | NA |
| Total | 224 | 142 | 64 | 18 | NA |
| B) Microscopy | | qPCR (all densities) | | | |
| | Total | Pf | Pv | Pf+Pv | Negative |
| Microscopy negative | 448 | 181 | 45 | 64 | 158 |
| Microscopy detects Pf | 131 | 83 | 7 | 40 | 1 |
| Microscopy detects Pv | 68 | 12 | 25 | 26 | 5 |
| Microscopy detects Pf+Pv | 12 | 3 | 2 | 7 | 0 |
| Total | 659 | 279 | 79 | 137 | 164 |
| | | qPCR (>20 parasites/μL) | | | |
| | Total | Pf (>20 parasites/μL) | Pv (>20 parasites/μL) | Pf+Pv (>20 parasites/μL) | Negative |
| Microscopy negative | 29 | 18 | 10 | 1 | NA |
| Microscopy detects Pf | 128 | 112 | 11 | 5 | NA |
| Microscopy detects Pv | 57 | 9 | 41 | 7 | NA |
| Microscopy detects Pf+Pv | 10 | 3 | 2 | 5 | NA |
| Total | 224 | 142 | 64 | 18 | NA |

In the top panels, infections positive by qPCR irrespective of density are included. In the bottom panels infections at densities >20 parasites/μL by qPCR are included. For the estimate for *P. falciparum*, infections at densities >20 *P. falciparum* parasites/μL by qPCR and <20 *P. vivax* parasites/μL are included, and vice-versa for the estimate for *P. vivax*.

## Capability of the miLab to determine parasite density

The miLab not only determines positivity but also parasite density. Correlation in density estimates between miLab and expert microscopy was assessed for samples from Ghana (n = 105). Correlation was high (Pearson's R = 0.742, n = 105, $P<0.0001$, with few outliers where miLab underestimated density by a large degree (Fig 2). On average density by miLab was 47% lower than density by expert microscopy.

## Discussion

Limited sensitivity of malaria diagnosis hampers case management and malaria control efforts. Diagnosis is particularly challenging if multiple parasite species co-occur in the same site, and where *hrp2/3* deletions are frequent, such as in northern Ethiopia. The sensitivity of the miLab digital microscope was higher than local microscopy conducted at the health center, and detected 94% of *P. falciparum* and 97% of *P. vivax* infections at densities above 200 parasites/

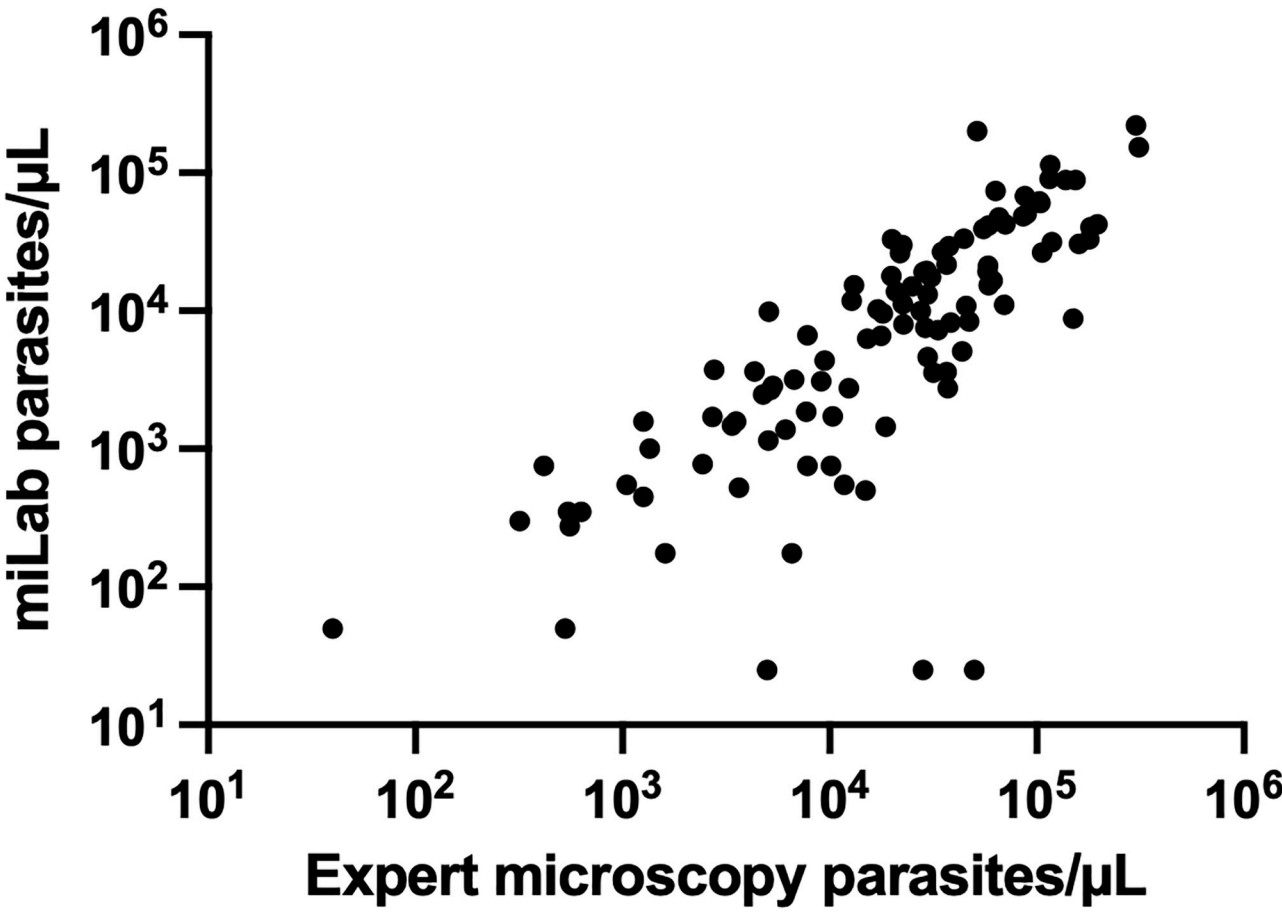

**Fig 2. Correlation between parasite quantification by miLab and expert microscopy for *P. falciparum*, B. samples where density estimates by microscopy differed by more than 2-fold, and samples with <50 parasites/µL by microscopy were excluded.**

µL. In Ethiopia, the miLab diagnosed virtually all infections with *hrp2* deletion, those infections could not be diagnosed by HRP2-based RDTs.

Ease of use is a key benefit of the miLab platform. While nucleic acid amplification tests such as PCR and LAMP are more sensitive and simplified protocols for use at point-of-care are increasingly developed, they all require basic laboratory skills to conduct multiple pipetting steps and interpret results [26–28]. Training of health center personnel to conduct these steps is a major challenge. In contrast, after the blood is directly loaded onto the miLab cartridge without any prior sample processing, all subsequent steps are conducted by the device. While in-depth usability studies are ongoing, the results from the current study suggest that the miLab can be reliably operated by health center personnel after minimal training. The ability to see the infected RBCs on the screen of the miLab to corroborate the diagnosis increases trust in the results.

While the overall sensitivity of the miLab was high, in Ethiopia in several *P. falciparum*-positive samples were misidentified as *P. vivax*, or vice-versa. Among samples positive by the miLab, it assigned the wrong species to 15/195 mono-infections, and identified only 5/18 mixed-species infections correctly. Accuracy of diagnosing the correct species was thus moderately better than by microscopy. Similar levels of misdiagnosis by microscopy as in this study were reported from other sites in Ethiopia [28]. Improvements in the miLab algorithm to

distinguish parasite species are ongoing, and the operator is able to verify the species when the result is displayed on the screen of the device. In the case of misclassification of species, any positive patient will receive treatment, though guidelines differ, with artemether-lumefantrine recommended for *P. falciparum*, and chloroquine for *P. vivax*. Either treatment is expected to clear either species in many cases, and thus incorrect species diagnosis might be considered a lesser problem than false-negative results where patients remain untreated. Yet, misdiagnosis increases the risk of treatment failure due to drug resistance [29].

A striking difference in specificity was observed between Ethiopia (specificity = 86%) and Ghana (specificity = 96%). This was surprising, as the use of the miLab prevents variation in slide quality, and the same algorithm was used for parasite detection. The relatively low number of only 164 true negative samples in Ethiopia might have reduced accuracy of specificity estimates in this site. Further, despite controls on data collection, eventual sample mix up at the health center cannot be fully ruled out.

The miLab allows quantification of parasite density. While the correlation between density by miLab and expert microscopy was moderate, it is comparable to correlations observed between expert microscopy and qPCR [30]. An approximately two-fold systematic underestimation of parasite density by the miLab was observed, this will be addressed in future improvements of the algorithm. The WHO has issued special recommendations for the treatment and monitoring of individuals with hyperparasitemia, i.e. individuals with more than 4% of RBCs infected, and *P. falciparum* parasitemia >10% is one of the factors defining severe malaria [31]. The miLab will enable health centers to identify these patients and ensure their proper treatment. Density data might also help to understand malaria epidemiology, and could be used for quality assurance of local microscopy.

In conclusion, the miLab is a valuable addition to the toolkit for malaria diagnosis. Its sensitivity is substantially better than field microscopy, and on par with RDT.

## Supporting information

**S1 Checklist. Inclusivity in global research.**
(DOCX)

**S2 Checklist.**
(DOCX)

**S1 Fig. Density by qPCR of samples screened for 500,000 RBCs.** Samples found positive by miLab are indicated in red.
(TIFF)

**S1 Data. Database.**
(XLSX)

## Acknowledgments

The authors thank all study participants and health centers personnel who supported this study. The authors appreciate the health directorate of Agona and Mankranso Government hospitals for their cooperation.

## Author Contributions

**Conceptualization:** Cristian Koepfli.

**Data curation:** Cristian Koepfli.

**Formal analysis:** Cristian Koepfli.

**Funding acquisition:** Cristian Koepfli.

**Investigation:** Yalemwork Ewnetu, Kingsley Badu, Lise Carlier, Claudia A. Vera-Arias, Emma V Troth, Abdul-Hakim Mutala, Stephen Opoku Afriyie, Thomas Kwame Addison, Nega Berhane, Wossenseged Lemma, Cristian Koepfli.

**Methodology:** Cristian Koepfli.

**Project administration:** Cristian Koepfli.

**Resources:** Kingsley Badu.

**Supervision:** Nega Berhane, Wossenseged Lemma, Cristian Koepfli.

**Writing – original draft:** Cristian Koepfli.

**Writing – review & editing:** Yalemwork Ewnetu, Kingsley Badu, Lise Carlier, Claudia A. Vera-Arias, Emma V Troth, Stephen Opoku Afriyie, Thomas Kwame Addison, Cristian Koepfli.

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
