## [Decision Letter · Decision Letter 0]

16 Jan 2024

PGPH-D-23-02507

A digital microscope for the diagnosis of Plasmodium falciparum and Plasmodium vivax, including P. falciparum with hrp2/hrp3 deletion

Dear Dr. Koepfli,

Thank you for submitting your manuscript to PLOS Global Public Health. After careful consideration, we feel that it has merit but does not fully meet PLOS Global Public Health’s publication criteria as it currently stands. Therefore, we invite you to submit a revised version of the manuscript that addresses the points raised during the review process.

Before the manuscript can be accepted for publication, the following 4 items raised by the reviewers need to be addressed:

As raised by reviewer 1, the Noul miLab appears to have substantial issues in accurately diagnosing mixed infections. This is particularly evident in the Ethiopian setting where P. vivax is prevalent, and could be a major shortcoming in other settings where the proportion of non-falciparum species is high or increasing. Particularly given that the authors disclose funding by Noul Inc. it is important that shortcomings such as the diagnosis of mixed infections are more prominent in the manuscript and given more discussion. Perhaps even bring to this forefront in the abstract. Please complete the STARD checklist and flow-chart (see reviewer 2 comment 1).Please provide further details on the definition of 'expert microscopy' and check for other gaps in the minimum reporting standards for malaria microscopy (see reviewer 2 comment 1 and reviewer 1 comment 6).Please provide feedback on why the specificity between Ghana and Ethiopia (even just for P. falciparum) was so substantial (see reviewer 2 comment 2).

We also highly recommend addressing the following points raised by the reviewers:

Please provide clarity on the hrp2/3 genotyping method (reviewer 1 comment 2)Please discuss the different RDTs used in Ghana and Ethiopia and how this may have impacted the results (reviewer 1 comment 5)Please add a comment on sample size calculation (reviewer 2 comment 3)

We look forward to receiving your revised manuscript.

Kind regards,

Sarah Auburn

Academic Editor

Journal Requirements:

2. Please include a complete copy of PLOS’ questionnaire on inclusivity in global research in your revised manuscript. Our policy for research in this area aims to improve transparency in the reporting of research performed outside of researchers’ own country or community. The policy applies to researchers who have travelled to a different country to conduct research, research with Indigenous populations or their lands, and research on cultural artefacts. The questionnaire can also be requested at the journal’s discretion for any other submissions, even if these conditions are not met.  Please find more information on the policy and a link to download a blank copy of the questionnaire here: https://journals.plos.org/globalpublichealth/s/best-practices-in-research-reporting. Please upload a completed version of your questionnaire as Supporting Information when you resubmit your manuscript.

3. Please send a completed 'Competing Interests' statement, including any COIs declared by your co-authors. If you have no competing interests to declare, please state "The authors have declared that no competing interests exist". Otherwise please declare all competing interests beginning with twhe statement "I have read the journal's policy and the authors of this manuscript have the following competing interests:"

4. Please amend your detailed Financial Disclosure statement. This is published with the article. It must therefore be completed in full sentences and contain the exact wording you wish to be published.

5. Please provide separate figure files in .tif or .eps format only and remove any figures embedded in your manuscript file. Please also ensure all files are under our size limit of 10MB.

Additional Editor Comments (if provided):

Reviewers' comments:

Reviewer's Responses to Questions

**Comments to the Author**

1. Does this manuscript meet PLOS Global Public Health’s publication criteria? Is the manuscript technically sound, and do the data support the conclusions? The manuscript must describe methodologically and ethically rigorous research with conclusions that are appropriately drawn based on the data presented.

Reviewer #1: Partly

Reviewer #2: Yes

2. Has the statistical analysis been performed appropriately and rigorously?

Reviewer #1: I don't know

Reviewer #2: Yes

3. Have the authors made all data underlying the findings in their manuscript fully available (please refer to the Data Availability Statement at the start of the manuscript PDF file)?

Reviewer #1: Yes

Reviewer #2: Yes

4. Is the manuscript presented in an intelligible fashion and written in standard English?

Reviewer #1: Yes

Reviewer #2: Yes

5. Review Comments to the Author

Reviewer #1: Koepfli et al reports ‘A digital microscope for the diagnosis of Plasmodium falciparum and Plasmodium vivax, including P. falciparum with hrp2/hrp3 deletion’. Authors report malaria diagnostic challenges especially where multiple Plasmodium species are endemic. To address such challenges, the performances of a digital automated portable digital microscope from Noul MiLab were evaluated among 659 suspected malaria patients in Gondar, Ethiopia and 991 patients in Ghana. According to authors report miLab shows superior performance as compared to microscopy and comparable performance with RDTs. Authors also concluded that miLab was more sensitive than microscopy and thus is a valuable addition to the toolkit for malaria diagnosis, particularly for areas with high frequencies of hrp2/3 deletions.

Comments

1. The details of the performance of digital microscopy for the diagnosis of falciparum malaria in Ethiopia and Ghana were not well presented.

2. Authors didn’t indicate which specific regions of the hrp2/3 loci (exon 2, exon 1-2, upstream or downstream flanking regions) were genotyped to determine hrp2/3 gene deletion.

3. It’s unclear whether Noul MiLab, MiLab and digital microscope are interchangeably used as each appears at various places.

4. The specificity of the miLab was 94.0%, which indicates the presence of ‘false positive’ results.

5. The same RDTs were not used in Ethiopia and Ghana, hence, comparison of the RDT sensitivity seems challenging between the 2 countries.

6. Under MM section (line 116-117), it reads that ‘All samples were tested on site by miLab, and RDT’. I’m wondering why microscope wasn’t used onsite for malaria diagnosis given the study site is only 2,100meters away from Gondar town of Ethiopia?. Is this for Ghana or Ethiopia? Does the national policy of Ethiopia support this?

7. In line 180-181, it reads that ‘If a sample was positive by qPCR for P. falciparum but the miLab diagnosed possible P. vivax (or vice versa), it was counted as positive’. In country like Ethiopia where Pf and Pv are the two plasmodium species that co-exist, species identification is critical as the country has species-specific treatment policy. Does this give sense as far as miLab-based malaria diagnosis is concerned in Ethiopia?

8. Under data analysis (line 182-183), it reads that ‘the number of infections where the incorrect species was diagnosed, or mixed-species infections with only one species diagnosed, are presented separately’. It’s unclear why such data analysis is considered and authors shall provide details. The data analysis has to be made for all the infections regardless of whether they are correctly diagnosed or not.

9. How authors did select the study sites in Ethiopia and Ghana?

10. Did authors enroll every febrile patient who sought malaria diagnosis at both health centers of Ethiopia and Ghana? How authors account for the variation in sensitivity of DBS-based qPCR (in Ethiopia) vs whole-blood based PCR (Ghana) for malaria diagnosis?

11. It is known that malaria is more endemic to Ghana than Ethiopia; however, it appears that the prevalence of malaria looks higher in Ethiopia (75.1% (495/659) individuals tested positive) than in Ghana (40.6% 402/991) tested positive) for P. falciparum. How authors justify this?

12. Line 209-212, under ‘Sensitivity and specificity of P. falciparum and P. vivax diagnosis’. Is it possible for the authors to calculate the combined sensitivity of miLab (94.3% at densities >200 parasites/µL) for both Ethiopia and Ghana given Pv is uncommon in the latter? Or is the sensitivity of miLab for Pf only?

13. Although the sensitivity and specificity of miLab were calculated as shown in table 1; it’s unclear which tool(s) (qPCR or microscopy) was taken as a gold standard. Or do authors think/believe that miLab is superior to qPCR?

14. It would be important to describe the total number of Pf and Pv samples subjected to calculate miLab sensitivity/specificity in table 1 so that readers can easily appreciate its diagnostic performances.

15. The specificity of microscope and RDT is superior to that of miLab (table 1) in Ethiopia, how authors justify that miLab could lead to the treatment of uninfected individuals with antimalarial drugs? i.e from drug pressure, or drug resistance emergence/development perspectives?

16. Line 241-245, it’s indicated that ‘the miLab successfully diagnosed 51/52 (98.1%) infections with hrp2 deletion while field microscopy diagnosed 37/52 (71.2%) of hrp2-negative infections; what justification could authors provide for the inability of the microscope to detect of hrp2-deleted parasites while miLab did? Authors shouldn’t underestimate the issue of Pv misdiagnosis (5 infections) in Ethiopia.

17. Line 238, when authors present ‘Diagnosis of hrp2/3 negative P. falciparum infections’, it would be imperative if they also provide ‘Diagnosis of hrp2/3 positive P. falciparum infections’ for readers to appreciate it.

18. Line 249-253, authors report that miLab has severe shortcomings in accurately diagnosing mixed infections (either the wrong species was assigned, or only one species was diagnosed in mixed species infections). Thus, in setting like Ethiopia where Pf and Pv are common and species-specific treatment policy exists, miLab diagnosis would lead to wrong drug regimens. Nevertheless, don’t authors think that diagnosing Pf as Pv and vice versa has severe consequences?

19. Table 2A, from a total of 142 Pf infections (qPCR (>20 parasites/μL)), miLab correctly identified 33 of them, but misclassify 6 of them as negative and 6 as Pv. The issue of false negative and misdiagnosis looks significant. From a total of 64 Pv infections confirmed by qPCR, miLab correctly detects 50 of them as Pv, but misclassify 5 of them as negative and 9 of them as Pf infections. Of the 18 mixed infection by qPCR, miLab correctly identified 27.8% (5/18) of them while 5 of them as Pv only and 5 as Pf only.

Reviewer #2: Ewnetu et al. report results from a study to assess the diagnostic performance of the Noul miLab, a digital microscope for the diagnosis of malaria from whole blood. The manuscript is clearly written and should be accepted for publication once the minor issues detailed below are resolved.

1. Standardisation in reporting: Given that this is a study to assess diagnostic performance, the STARD checklist & flow-chart (https://www.equator-network.org/reporting-guidelines/stard/; BMJ 2015;351:h5

---

## [Decision Letter · Decision Letter 1]

21 Mar 2024

A digital microscope for the diagnosis of Plasmodium falciparum and Plasmodium vivax, including P. falciparum with hrp2/hrp3 deletion

PGPH-D-23-02507R1

Dear Mr. Koepfli,

We are pleased to inform you that your manuscript 'A digital microscope for the diagnosis of Plasmodium falciparum and Plasmodium vivax, including P. falciparum with hrp2/hrp3 deletion' has been provisionally accepted for publication in PLOS Global Public Health.

Best regards,

Sarah Auburn

Academic Editor

Reviewer Comments (if any, and for reference):

Reviewer's Responses to Questions

**Comments to the Author**

1. If the authors have adequately addressed your comments raised in a previous round of review and you feel that this manuscript is now acceptable for publication, you may indicate that here to bypass the “Comments to the Author” section, enter your conflict of interest statement in the “Confidential to Editor” section, and submit your "Accept" recommendation.

Reviewer #2: All comments have been addressed

2. Does this manuscript meet PLOS Global Public Health’s publication criteria? Is the manuscript technically sound, and do the data support the conclusions? The manuscript must describe methodologically and ethically rigorous research with conclusions that are appropriately drawn based on the data presented.

Reviewer #2: Yes

3. Has the statistical analysis been performed appropriately and rigorously?

Reviewer #2: Yes

4. Have the authors made all data underlying the findings in their manuscript fully available (please refer to the Data Availability Statement at the start of the manuscript PDF file)?

Reviewer #2: Yes

5. Is the manuscript presented in an intelligible fashion and written in standard English?

Reviewer #2: Yes

6. Review Comments to the Author

Reviewer #2: Comments adequately addressed.

7. PLOS authors have the option to publish the peer review history of their article (what does this mean?). If published, this will include your full peer review and any attached files.

**Do you want your identity to be public for this peer review?** For information about this choice, including consent withdrawal, please see our Privacy Policy.

Reviewer #2: No
